# Therapeutic Treatments for Osteoporosis—Which Combination of Pills Is the Best among the Bad?

**DOI:** 10.3390/ijms23031393

**Published:** 2022-01-26

**Authors:** Christian Horst Tonk, Sarah Hani Shoushrah, Patrick Babczyk, Basma El Khaldi-Hansen, Margit Schulze, Monika Herten, Edda Tobiasch

**Affiliations:** 1Department of Natural Sciences, Bonn-Rhein-Sieg University of Applied Sciences, von-Liebig-Str. 20, 53359 Rheinbach, Germany; christian.tonk@h-brs.de (C.H.T.); sarah.shoushrah@h-brs.de (S.H.S.); patrick.babczyk@h-brs.de (P.B.); basma.hansen@gmail.com (B.E.K.-H.); margit.schulze@h-brs.de (M.S.); Edda.Tobiasch@h-brs.de (E.T.); 2Department of Trauma, Hand and Reconstructive Surgery, University Hospital Essen, University of Duisburg-Essen, 45147 Essen, Germany

**Keywords:** osteoporosis, osteoblast, osteoclast, treatment, anabolic, catabolic, combination of treatments, bone remodeling, bone mineral density, biomaterial

## Abstract

Osteoporosis is a chronical, systemic skeletal disorder characterized by an increase in bone resorption, which leads to reduced bone density. The reduction in bone mineral density and therefore low bone mass results in an increased risk of fractures. Osteoporosis is caused by an imbalance in the normally strictly regulated bone homeostasis. This imbalance is caused by overactive bone-resorbing osteoclasts, while bone-synthesizing osteoblasts do not compensate for this. In this review, the mechanism is presented, underlined by in vitro and animal models to investigate this imbalance as well as the current status of clinical trials. Furthermore, new therapeutic strategies for osteoporosis are presented, such as anabolic treatments and catabolic treatments and treatments using biomaterials and biomolecules. Another focus is on new combination therapies with multiple drugs which are currently considered more beneficial for the treatment of osteoporosis than monotherapies. Taken together, this review starts with an overview and ends with the newest approaches for osteoporosis therapies and a future perspective not presented so far.

## 1. Bone Remodeling Process in Healthy Individuals

Bone is a rigid but metabolically active tissue that is vital for various physiological processes including mineral storage and homeostasis and endocrine functions [1,2]. On the tissue level, bone is composed of different layers including the periosteum, osseous tissues, the endosteum, and the bone marrow [3]. The periosteum is the outermost layer of bone and is composed of two layers: an outer fibrous layer and an inner osteogenic layer containing osteoprogenitor cells [4]. Because it houses these osteoprogenitor cells, the periosteum is vital for bone growth, remodeling, and fracture repair [1]. Furthermore, stem cells located in the periosteum were found to be more sensitive to growth factors than bone-marrow-derived stem cells, thereby signifying their role in bone regeneration after injury [2,5]. The endosteum is a very thin membrane that lines the inner surface of the wall of the bone marrow cavity [6]. The endosteum aids in bone repair and remodeling because, like the periosteum, it also contains osteoprogenitor cells. Additionally, by resorbing unnecessary osseous tissue from the bone cavity, the endosteum maintains the weight-to-strength ratio of long bones [3,7]. Bone marrow is a soft tissue that contains hematopoietic stem cells responsible for the regeneration of blood cells, and it also hosts mesenchymal stem cells (BM-MSCs), which are important for bone formation and repair [8,9]. Together, the complex cells of these layers work to generate and, in the case of injury, regenerate bone tissue.

Bone development begins early during gestation and continues into the first two decades of life [10]. It occurs through two processes: intramembranous ossification, which involves the formation of osteoblasts from MSCs, and endochondral ossification, where the cartilage extracellular matrix (ECM) produced by hypertrophic chondrocytes is replaced by bone [11]. Bone is maintained through the processes of modeling and remodeling. Bone modeling is the process in which bone is interchangeably formed by osteoblasts and resorbed by osteoclasts. It aims to increase bone mass and maintain or alter bone structure and is initiated due to local tissue strain. Contrary to bone modeling, during bone remodeling, the processes of bone formation and resorption occur concurrently. Bone remodeling renews the bone to repair microdamage, and it is important for mineral homeostasis [12].

The processes of bone formation, growth, modeling, and remodeling involve the dynamic coordination of multiple progenitor and mature bone cells, including osteoblasts and osteoclasts. In the following sections, the origin of these osteoblasts and osteoclasts as well as their functions is discussed.

### 1.1. Bone Development: Differentiation and Activation of Osteoblasts

Osteoblasts (OBs) originate from mesenchymal stem cells and are responsible for bone formation or osteogenesis [13,14,15,16,17]. In bone tissue, MSCs can originate from the adjacent area to vessel walls, the periosteum, and the endosteum [18,19,20,21,22]. The formation of osteoblasts from MSCs occurs through multiple stages, which can be divided into proliferation, ECM deposition, mineralization, and apoptosis. The differentiation and development of osteoblast can be characterized by the expression of multiple osteoblastic markers such as alkaline phosphatase (ALP) and alpha-1 type I collagen (COL1A1) mainly in the early stages of differentiation and osteocalcin (OCN), osteopontin (OPN), and bone sialoprotein (BSP) at later stages of differentiation [23,24,25]. The differentiation of MSCs starts by the commitment of MSCs towards a specific cell lineage followed by differentiation [26,27,28]. This commitment and subsequent differentiation are controlled by specific transcription factors; in osteoblastogenic differentiation, the runt-related transcription factor 2 (RUNX2) is a master regulator [29,30]. RUNX2 expression has been shown to induce the differentiation of MSCs into pre-osteoblasts and inhibit the commitment towards the adipocyte lineage [28]. Studies have shown that RUNX2 expression is upregulated in pre-osteoblasts. It reaches its highest expression level in immature osteoblasts, and after that, its expression is downregulated in mature osteoblasts [30,31,32]. RUNX2 regulates the expression of genes that are important for bone formation such as COL1A1, ALP, BSP, and OCN [5]. RUNX2 additionally enhances the expression of osterix (OSX), which is responsible for the further differentiation of OBs and maturation by the activation of ALP and subsequent mineralization [33,34]. RUNX1 is highly expressed in osteoblasts and was also recently found to be an important regulator of osteogenesis [35,36,37,38]. RUNX1 is important for lineage commitment of bone marrow MSCs (BM-MSCs), as shown in the study by Luo and colleagues, where the knockdown of RUNX1 decreased the osteogenic capacity of BM-MSCs in favor of an increased adipogenic capacity [35]. They showed that RUNX1 promotes the osteogenic capacity through the canonical Wnt/β-catenin pathway [35]. Tang and colleagues reported a similar function of RUNX1. They showed that RUNX1 was important for signaling in chondrocytes towards an osteoblast lineage commitment and thus promoting endochondral bone formation in a murine model [36]. RUNX1 was also shown to upregulate the expression of the transcription factors RUNX2 and activating transcription factor 4 (ATF4), as well as other genes involved in osteogenesis and bone homeostasis such as OCN or OSX [37]. Recently, Tang and colleagues reported that RUNX1 improves osteogenesis through the upregulation of the WNT/β-catenin signaling pathway and other genes such as bone morphogenetic protein 7 (BMP-7) and bone morphogenetic protein receptor, type IA [38]. Other transcription factors also promote osteoblastogenesis such as distal-less homeobox 5 and ATF4 through their interactions with RUNX1 and RUNX2 [39,40]. The formation of OBs is governed by signaling pathways, including the transforming growth factor-β (TGF-β) superfamily and their cross-talk with BMP signaling, as well as Wnt and the hedgehog signaling pathways [41]. The Wnt/β-catenin pathway is known for its role in promoting osteoblastogenesis through the binding of Wnt ligand and the subsequent downstream translocation of β-catenin followed by the expression of osteoblastic genes [42]. Non-canonical Wnt signaling is characterized by activation through phosphorylation cascades when specific receptors are activated by specific ligands such as Wnt5a or Wnt11, resulting in an increase in intracellular Ca^2+^ levels which lead to a signaling cascade that results in the activation of RUNX2 [42,43]. Another important signaling pathway is BMP signaling. Of the known BMPs, BMP-2, 4, 5, 6, 7, and 9 are known to have a positive effect on osteogenesis [44]. BMP-2 and BMP-7 have been approved for clinical use in bone regeneration by the Food and Drug Administration (FDA) [45].

When they are fully differentiated, OBs start producing bone matrix [46]. This process occurs in two stages: the deposition of the organic matrix (osteoid) followed by its mineralization [2,5]. The first stage entails the secretions of collagens (mainly type I), non-collagen proteins (osteonectin (ON), OPN, BSP, and OCN), and proteoglycans. The mineralization of the osteoid is achieved by the deposition of minerals such as hydroxyapatites. After this, the OBs either become osteocytes, bone lining cells, or go into apoptosis [2,5].

In contrast to bone-synthesizing osteoblasts, bone-degrading osteoclasts play a crucial role during bone remodeling.

### 1.2. Bone Autophagy: Differentiation and Activation of Osteoclasts

Osteoclasts (OCs) are the cells responsible for bone resorption, a process where the minerals of the bone are dissolved and the organic matrix is degraded [47]. Osteoclasts originate from hematopoietic stem cells, specifically from monocyte–macrophage progenitors, and are terminally differentiated cells [2]. Mature osteoclasts are dome-shaped, multinucleated cells. The multinucleation is a hallmark feature of osteoclasts, and they have on average eight nuclei [48].

Hematopoietic precursors that form osteoclasts are recruited from peripheral blood or bone marrow. The recruitment and generation of osteoclasts is controlled by several factors such as cytokines derived from the osteoblasts and osteoclast, and calcium gradients [16]. The mononucleated OC precursor’s commitment is stimulated in response to interleukin-3 (IL-3), granulocyte-macrophage colony-stimulating factor (GM-CSF), and macrophage colony-stimulating factor (M-CSF) [2,49]. Then, the committed mononucleated preosteoclasts fuse under the influence of receptor activator of nuclear factor kappa-Β ligand (RANKL), forming the mature multinucleated osteoclast [49]. The cross-talk between the bone cells is important for the process of osteoclastogenesis. For instance, M-CSF is secreted by mesenchymal osteoprogenitor cells and osteoblasts, and RANKL is secreted by osteoblasts and osteocytes [5]. The activation of the interaction between RANKL and its receptor RANK activates the expression of osteoclast-specific genes such as tartrate-resistant acid phosphatase (TRAP) and cathepsin K, which are vital proteins for OC activity [5,50]. The secretion of RANKL by OBs is regulated by vitamins and cytokines such as vitamin D, parathyroid hormone (PTH), IL-1 and IL-6 [2,51]. Osteoprotegerin (OPG) is also produced by Obs; however, it prevents RANKL/RANK interaction by binding to RANKL, thus inhibiting osteoclastogenesis [50,52].

When osteoclasts mature, the process of bone resorption starts by their polarization forming an apical membrane domain that interacts with the bone surface and an opposing basolateral membrane domain located away from bone, giving it its dome shape. The apical membrane forms ruffled borders and the sealing zone where resorption is initiated [53]. At the ruffled apical border, the vacuolar-type H+-ATPase acidifies the resorption lacuna, causing an acidic environment through the secretion of protons, resulting in the dissolution of hydroxyapatite crystals and minerals in the matrix [2,5]. The exposed osteoid is then digested in response to the decrease in pH by secreted enzymes, such as cathepsin K and matrix metalloproteinases (MMPs) (MMP-14 and MMP-9) [2,5]. TRAP is also secreted by OCs, although the exact physiologic role of it remains unclear [2,5]. When the matrix is degraded in the sealed zone, the degraded products are endocytosed and released into the ECM through the basolateral membrane [54]. The number and life span of osteoclasts impacts the amount of bone resorption. After carrying out bone resorption, osteoclasts undergo apoptosis. Estrogens and androgens inhibit osteoclast generation while improving osteoblast’s survival and bone mineralization [2,55]. Moreover, estrogens indirectly induce apoptosis in osteoclasts. Therefore, a decrease in estrogen levels during and after menopause is the main cause of bone loss and osteoporosis [5,56]. As estrogen is taken up by osteoblasts and has an indirect influence on osteoclasts, it directly influences the interplay between these two cell types. In the following chapter, the feed-back loop of osteoblasts and osteoclasts is described in more detail.

### 1.3. Interplay of the Major Cell Types Involved in Bone Remodeling: Osteoblasts and Osteoclasts

Bone modeling and remodeling are important processes that maintain bone structural integrity and homeostasis (Figure 1). This in turn requires the careful orchestration and organization of osteoblastic and osteoclastic activities, which are achieved by the regulated cross-talk between osteoclasts and osteoblasts. Bone remodeling occurs in four different stages, which are activation, resorption, reversal, and formation [5,12]. These stages take place in bone cavities where basic multicellular units (BMUs) are found. BMUs consist of bone-resorbing OCs, which are in the front, creating the cutting cone, and OBs behind them create the closing cone, with both being connected and interacting via blood vessels [5,12]. The first stage is recruitment and maturation of OCs followed by bone resorption. In the reversal stage, OCs cease their activity and undergo apoptosis while OBs are recruited, followed by formation of the osteoid and mineralization of bone matrix by OBs [12,57]. The cross-talk between OCs and OBs can be through direct or indirect interactions. Direct cell-to-cell interaction between OBs and OCs is effectuated through membrane-bound mediators such as semaphorins (semaphorin 3a), FAS ligands or ephrins (ephirinB2) [58]. Estrogen induces the apoptosis of OCs, and this process was found to be regulated through FAS ligand (FASL)/FAS interaction [59]. This occurs when estrogen induces an upregulation of FASL expression in osteoblasts, resulting in the apoptosis of pre-osteoclasts [60]. Additionally, matrix metalloproteinase-3 (MMP-3) was found to be upregulated by estrogen and was responsible for the cleavage and solubilization of FASL, which in turn induced osteoclast apoptosis [61]. The cross-talk could also be indirect in the case of soluble factors released of each cell type such as M-CSF, OPG, RANKL, WNT5A, and WNT16 from osteoblasts and sphingosine 1 phosphate, semaphorin 4D, and complement component C from the osteoclasts [62]. Additionally, the release of miRNAs and exosomes from the cells as well as the matrix-derived coupling factors released by the resorption of the mineralized matrix play a role, which should not be underestimated [58,62]. As mentioned previously, osteoblasts and mesenchymal progenitors produce M-CSF, which is vital for osteoclastogenesis and was found to upregulate the expression of RANK in the hematopoietic precursor cells [49,63]. In addition, OBs release RANKL and OPG, which can either induce or inhibit osteoclastogenesis [64]. Additionally, complement component 3a was isolated in vitro from a co-culture system from OCs’ conditioned media and was found to improve OBs’ activity by improving ALPs’ mineralization [65].

Osteocytes also have a role in regulating bone remodeling [66]. Osteocytes found within the mineralized matrix act as mechanosensors coordinating bone remodeling by controlling osteoblasts and osteoclasts activities [67]. When subjected to mechanical or hormonal signals (such as circulating PTH), osteocytes release factors such as OPG, RANKL, and sclerostin that affect osteoclastic and osteoblastic activities [68,69,70]. Sclerostin, a protein that in humans is encoded by the SOST gene, was found to decrease bone formation and improve bone resorption [67,70,71,72,73]. Osteocytes also have the ability to change their microenvironment by the process of osteocytic osteolysis, or perilacunar remodeling (PLR) [74,75]. Lactation-induced PLR maintains mineral homeostasis and the maternal skeleton’s load-bearing capacity [76]. The osteolytic activities of osteocytes were shown to be accompanied by an increase in osteoclast gene expression in osteocytes such as TRAP, carbonic anhydrase 2, and cathepsin K and that sclerostin could induce the osteolytic activities by inducing them [72].

Considering the numerous factors in bone remodeling, it is no surprise that dysregulation in the balance between the osteoclastic and osteoblastic activities can lead to bone diseases where the bone mass is increased or decreased. Osteoporosis is a condition where the balance between resorption and formation is disturbed, and increased resorption results in decreased bone density and a higher risk for bone fractures [5]. Conversely, osteopetrosis is a rare genetic disease, in which mutations lead to decreased bone resorption, causing the uneven accumulation of bone mass and thus too dense bones [5]. Paget’s disease is the second most common example of diseases that occur due to the dysregulation of bone remodeling where, similar to osteoporosis, bone absorption is increased and in this case accompanied by disorganized bone formation [77].

## 2. Changes in the Bone-Remodeling Process in Osteoporosis

Osteoporosis is a chronical, systemic skeletal disorder characterized by an increase in bone resorption which leads to reduced bone density. As one of the major age-related diseases, osteoporosis (OP) is caused by the imbalance between bone formation and bone resorption. According to the pathogenesis and causes, OP can be divided into two major types, namely primary and secondary OP. The major types of osteoporosis in humans are: postmenopausal osteoporosis (primary OP of type I), disuse osteoporosis, (primary OP of type II with advancing age/senile OP), and adverse reaction to long-term medication for the treatment of diseases (secondary OP, glucocorticoid-induced osteoporosis). Osteoporosis is a chronical, systemic skeletal disorder characterized by an increase in bone resorption, which leads to reduced bone density. The reduction in bone mineral density (BMD) and therefore low bone mass result in an increased fracture risk [78]. The International Osteoporosis Foundation and European Society for Clinical and Economic Aspects of Osteoporosis and Osteoarthritis published a guide for the diagnosis of osteoporosis [79]. The main clinical assessment of osteoporosis is the T-score for BMD measured with dual-energy X-ray absorptiometry (DXA or DEXA), a technology that measures the BMD of the cortical and trabecular areal. Bone loss due to a low BMD (T-score below 2.5 SD) occurs among all genders, but women aged 50 years or older have a prevalence for osteoporosis four times higher than that of men [80]. An evaluation of the most recently discovered molecular mechanisms in osteoporosis can help to uncover a better understanding of its etiology and preference for elderly women. As described earlier, estrogen is the key hormone that regulates bone density and maintains the equilibrium between bone formation and bone resorption by either enhancing the proliferation of osteoblasts or by diminishing the levels of osteoclasts [81]. During menopause or after a surgical removal of the ovaries, estrogen levels decrease rapidly [82,83]. Furthermore, the serum estradiol concentration decreases up to 90% and the serum concentration of estrone, a weak estrogen and minor female sex hormone, decreases up to 75% [84]. Estrogen and its derivates in general are described as critical factors for skeletal homeostasis as well as bone remodeling. It acts through two receptors, estrogen receptor-alpha (ER-α) and estrogen receptor-beta (ER-β). ER-α was described to be especially important for bone regulation [85,86,87,88]. Furthermore, estrogen modulates the OPG/RANKL system [89]. Streicher and colleagues were able to track the increased bone resorption in osteoporosis down to the lack of ER-α-mediated suppression of RANKL expression in osteoblasts. The lack of estrogen therefore leads to an increased RANKL expression, which causes the activation, differentiation, and survival of bone-resorbing osteoclasts, which tend to an overactivity of osteoclasts and to an increased bone loss [90]. Osteoprotegerin, on the other hand, works as a decoy receptor and binds RANKL. Thus, RANKL is inhibited and thus, the RANK–RANKL interaction which inhibits osteoclastogenesis and activation is affected. Since the expression of RANKL in osteoblasts is constantly activated, when estrogen is missing, osteoclasts will also become constantly activated and differentiated, thereby leading to overactivity, causing increased bone resorption [90]. Estrogen also has important direct effects on osteoblasts by promoting the differentiation of MSCs towards the osteogenic lineage and by increasing the production of growth factors, such as insulin-like growth factor 1 (IGF-1) and TGF- β [91,92]. Therefore, the loss of estrogen in osteoporotic patients leads to a decreased differentiation of osteoblasts and growth factor production. Additionally, estrogen was described to have a suppressive effect on the Wnt-signaling antagonist sclerostin, which is lacking in osteoporosis and leads to the suppression of Wnt signaling by active sclerostin [93]. Therefore, estrogen affects the skeletal anabolism and homeostasis, but if estrogen is missing, the catabolism is promoted and bone homeostasis cannot be ensured anymore [94].

In addition to the sex steroid estrogen, other hormones change with aging and may contribute to the development of osteoporosis, namely the diminishing production of IGF-1 and IGF-2 by the liver, which leads to a decreased differentiation and activation of osteoblasts and therefore promotes osteoporosis [95]. Besides the decreased concentration of IGFs, an increased concentration of the inhibitory IGF binding protein (IGFBP) was observed [96]. On the contrary, Ye and colleagues showed that IGFBP7 treatment in an ovariectomy-induced osteoporosis mouse model attenuated osteoporotic bone loss by inhibiting the activity of osteoclasts and therefore suppressed osteoporosis [97].

Other risk factors that can contribute to secondary osteoporosis include medical disorders, medications, poor nutrition or dietary factors, and the lifestyle choices of patients (Table 1). The number of risk factors can increase the chances of developing osteoporosis. Patients with the following medical problems have a higher risk for developing osteoporosis: cancer, in particular breast cancer [98,99], rheumatoid disorders such as rheumatoid arthritis [100,101] or systemic lupus erythematosus [102], chronic kidney or liver diseases [103,104], diabetes mellitus [105,106], Parkinson’s disease [107], and multiple myeloma [108]. Eating disorders such as anorexia [109] nervosa, poor nutrition, or dietary factors such as a low calcium intake [110] can also cause osteoporosis, as calcium plays an important role in the development of bone. The lack of calcium or a low intake contribute to diminished bone density and early bone loss, which lead to an increased fracture risk [111]. Lifestyle choices are important, too; a sedentary lifestyle with sitting most of the time leads to a higher risk of osteoporosis [112]. Additionally, excessive alcohol consumption [113,114] and the use of tobacco [113,115,116] can increase the risk of osteoporosis.

## 3. State of the Art Treatments against Osteoporosis

Osteoporosis can be caused by a variety of risk factors, and therefore, many different treatments exist. Precaution to lower the risks for osteoporosis or intervention for those already suffering from osteoporosis can be achieved by respective lifestyle choices. A healthy and varied diet with fresh fruits, vegetables, and calcium-rich and vitamin-rich food (vitamin D/vitamin K/vitamin E) is described to be the basis for osteoporosis prevention and therapy [117]. Actual ongoing studies investigate the influence of probiotica (i.e., *Lactobacillus reuteri*) (NCT04169789) or modifiers of the gut flora (blackcurrant extract) (NCT04431960) for the prevention of OP. Limiting the consumption of alcohol as well as avoiding smoking can also help to reduce the risk of developing osteoporosis [113]. Regular exercise and physical activity help to treat osteoporosis, but different strategies differ in their impact. Weight-bearing aerobic activities as well as resistance training are most suitable for osteoporotic patients, but also, flexibility and stability exercises as well as endurance sports can improve the status of the disease [118,119]. Hereby, physical activity is not only beneficial for the stimulation of bone tissue metabolism, which results in the gain and maintenance of bone mass, but also for other factors such as muscle strength or body balance to prevent falls and fractures [120].

However, all these lifestyle approaches do not work alone to prevent or to treat osteoporosis. Therefore, drug-based medication is necessary, and additionally, alternative medical treatments involving the consumption of certain herbs were described [121]. A variety of medications are currently used which differ in their functional mechanisms and activity (Figure 2). A majority of drug-based medications can be categorized in anabolic treatments, which activate bone-synthesizing osteoblasts [122], and catabolic treatments, which inhibit excessive bone degradation by osteoclasts [123]. The most commonly used anabolic treatments are parathyroid hormones, such as teriparatide and abaloparatide, and monoclonal sclerostin antibodies, such as Romosozumab. The most commonly used catabolic treatments are bisphosphonates, namely alendronate and zoledronate, selective estrogen receptor modulators, such as tamoxifen and raloxifene, and monoclonal RANKL antibodies, such as Denosumab. Novel and cutting-edge approaches to treat osteoporosis include the combination of anabolic and catabolic treatments.

### 3.1. Anabolic Treatments of Osteoblasts to Improve Bone Growth

Anabolic treatments of osteoporosis came more into focus since bisphosphonate therapy was the standard of care for the prevention and treatment of glucocorticoid-induced osteoporosis. These patients were assessed to have a high risk for fractures, as the normal bone turnover depending on the balance between osteoblast and osteoclast activity is disturbed by the drug. Glucocorticoids can cause rapid bone loss, increasing bone resorption and decreasing bone formation via apoptosis of the osteoblasts. Anabolic treatment (osteoanabolic therapy) is a promising therapeutic strategy against such an outcome, and the most promising anabolic drugs are described in the following chapters.

#### 3.1.1. Parathyroid Hormones and Hormone-Related Bone Growth Agents Increase Osteoblast Activity

Parathyroid hormone (PTH) is the hypercalcemic hormone of the body, and when plasma calcium levels are decreased, PTH acts on the kidney, bone, and/or small intestine to increase plasma calcium. PTH is secreted primarily by the chief cells of the parathyroid glands. The polypeptide contains 84 amino acids. Its action is opposed by the hormone calcitonin [124].

The parathyroid-hormone-related protein (PTHrP) is a protein member of the parathyroid hormone family. PTHrP acts as an endocrine, autocrine, paracrine, and intracrine hormone. It regulates endochondral bone development by maintaining the endochondral growth plate at a constant width. The protein is secreted by mesenchymal stem cells. It is occasionally also secreted by cancer cells such as breast cancer, and certain types of lung cancer, including squamous-cell lung carcinoma [125].

Two types of PTH receptors are known. Parathyroid hormone 1 receptor (PTH1R) is activated by the 34 N-terminal amino acids of PTH and by PTHrP, being present at high levels on cells of bone and kidney, and parathyroid hormone 2 receptor (PTH2R) is present at high levels on cells of central nervous system, pancreas, testes, and placenta. Since PTH influences bone remodeling and is secreted in response to low blood serum calcium levels, it indirectly stimulates osteoclast activity within the bone matrix, in an effort to release more ionic calcium into the blood to elevate a low serum calcium level through adenylate cyclases and phospholipase C. The activated receptor leads to increased RANKL expression, which binds to RANK on osteoclasts, which activates osteoclasts to ultimately increase the resorption rate [126]. It was shown that in the osteocyte, PTH regulates RANKL expression through the inhibition of salt-inducible kinases (SIKs) and the nuclear translocation of cyclic adenosine monophosphate-regulated transcriptional coactivator, CREB-regulated transcription coactivator 2, which is a known substrate of SIKs. It was also reported that PTH-induced SIK inhibition allows for the nuclear translocation of the histone deacetylases (HDACs) 4 and 5, which inhibit the transcription factor myocyte enhancer factor 2C and therefore decrease Sost gene expression, which is a negative regulator of bone formation [127,128].

In the first study on PTH in 1983, it was revealed that PTH (fragment 1-34; PTH 1-34) increases bone formation and resorption in dogs [129]. Later, it was shown that the effect of different concentrations of PTH 1-34, ranging between 5 and 80 µg/kg body weight, decreased the risk of fractures in osteoporosis patients [130,131]. There are now two analogs used to increase bone formation as a treatment for osteoporosis: teriparatide and abaloparatide [132]. Teriparatide (recombinant human PTH; TPTD) is the active form of PTH 1-34 and showed stimulation of bone formation by increasing osteoblast numbers [133], enhanced osteoblast differentiation [134], and increased bone mineral density [135,136]. During the first decade of the 2000s, it was the only FDA-approved drug, which could replace bone lost due to osteoporosis. The activation of PTH1R activates multiple signaling pathways. Cupp and colleagues showed that both N-terminal and C-terminal domains of PTH and PTHrP are critical for the activation [137]. Additional studies also revealed a promising role of TPTD in osteogenesis imperfecta [138]. Due to the influence on the OPG/RANKL system, resulting in osteoblast activation and increased bone formation, teriparatide can have an impact on patients with osteogenesis imperfecta [139].

The second FDA-approved drug for osteoporosis treatment (osteoanabolic) is abaloparatide (ABL), which is an analog of parathyroid-hormone-related protein (PTHrP 1-36). Compared to TPTD, abaloparatide showed higher bone mineral density, but bone formation was not increased in clinical studies [140]. The administrated concentration of ABL was between 20 and 80 µg/kg body weight, similar to teriparatide [141]. Both drugs increased bone formation marker procollagen type 1 N-terminal propeptide in serum with little effect on bone resorption marker DPD/Cr in urine, suggesting anabolic effects. In in vitro studies, Ricarte and colleagues showed that the peptides (PTH 1-34; PTHrP 1-36, ABL) modulate osteoblastic RANKL expression [142] and the group of Makino reported a similar expression of bone-related factors, IGF-1 and osteocalcin, and no differences in the effect of Wnt signaling inhibitors such as sclerostin [140]. Although ABL had no significantly improved effect, it was well tolerated for women with postmenopausal osteoporosis at high risk of fracture [143]. Clinical III phase studies showed that ABL was associated with increased heart rate and small decreases in blood pressure in postmenopausal women with osteoporosis, but no increased risks of serious cardiac adverse events, major adverse cardiovascular events, and heart failures [144]. In 2019, it was reported that the supporting effect of ABL in (mouse) chondrogenesis of mesenchymal stem cells was due to the inhibition of reactive oxygen species (ROS) production [145].

Taken together, both FDA-approved osteoanabolic drugs, TPTD and ABL, showed an increase in bone formation and a decreased risk of fractures in postmenstrual women with osteoporosis. Although the treatment is usually limited to two years, the positive effects of such an anabolic treatment could already be observed after six or 12 months. Actual ongoing studies investigate new application forms of PTH (injection vs. skin plaster) (NCT01674621) and therapy duration of PTH treatment (NCT03702140).

#### 3.1.2. Monoclonal Sclerostin Antibodies Promote Wnt Signaling in Osteoblasts

Antibodies against sclerostin, a protein produced by osteocytes and inhibiting bone formation by blocking the Wnt signaling pathway and thus inhibiting osteoblast activity, come more into focus to treat osteoporosis [146,147]. Sclerostin naturally binds to the LDL-receptor-related protein (LRP)-5 or 6-receptor on osteoblasts. When it is blocked, Wnt binds to the LRP5 or LRP6 coreceptors, which are specific receptors of the Frizzled family, leading to the activation of the WNT signaling pathway and bone formation. A prominent antibody against sclerostin, Romosozumab, was investigated in several studies. The working group of van Dinther could identify LRP6 as the main sclerostin receptor [148]. Additionally, the application of Romosozumab was associated with increased bone mineral density and bone formation in postmenopausal women with low bone mass [73,149]. The monthly subcutaneous administration of 70–210 mg showed a significant increase in bone mass [147]. Additionally, Romosozumab was associated with a lower risk of vertebra fractures [150,151]. However, different investigations suggest that additional bisphosphonates administration is necessary to maintain bone mass [152].

### 3.2. Catabolic Treatments of Osteoclasts to Prevent Excessive Bone Resorption

The catabolic treatment of osteoporosis aims to prevent excessive bone degradation of osteoclasts by different strategies. In this chapter, the most commonly used drugs which have an effect on osteoclast activity are introduced, discussed, and compared. This includes not only the drugs that are approved and currently prescribed, but also those that are promising as potential alternatives.

#### 3.2.1. Bisphosphonates Advance Osteoclasts toward Apoptosis

At the moment, bisphosphonates are the most commonly prescribed drugs to treat osteoporosis. In general, they possess two key therapeutic properties, which are the ability to bind to bone matrix and the inhibitory effect on osteoclasts. There are two types of bisphosphonates, namely non-nitrogen-containing and nitrogen-containing bisphosphonates. The first group contains the approved drugs etidronate [153], clodronate [154], and tiludronate [155], while the nitrogen-containing group consists of the approved drugs alendronate [156,157,158], risedronate [159,160], neridronate [161,162], ibandronate [163,164], olpadronate [165], pamidronate [166], and zoledronic acid or zoledronate (ZOL) [167]. The molecular mechanism of the nitrogen and non-nitrogen drugs is different but leads to the same result: osteoclasts will be led into apoptosis, resulting in a decreased bone resorption and an increased bone density. Bisphosphonates are remarkably structurally similar to pyrophosphate but differ by having a phosphorus–carbon–phosphorus bond instead of a phosphorus–oxygen–phosphorus bond [168]. Thus, bisphosphonates are more resistant to hydrolysis and highly resistant to degradation. The nitrogen-containing bisphosphonates have an additional nitrogen atom, while the non-nitrogen-containing bisphosphonates do not [168]. Nitrogen-containing bisphosphonates inhibit farnesyl pyrophosphatases (FPPs) in osteoclasts, which is an enzyme essential for their survival and function [169]. The non-nitrogen-containing bisphosphonates are metabolized by osteoclasts as substrate for adenosine triphosphate (ATP) after their uptake [170]. They will be used as terminal pyrophosphate in ATP, therefore resulting in an inactive form of ATP. This inevitably leads to an induction of apoptosis in osteoclasts [170]. Bisphosphonates vary in their affinity to bone, which determines their duration of action and their potency [169,171]. Based on those properties, bisphosphonates can be taken orally or intravenously either weekly, monthly, or even yearly. Currently, the most potent approved drug for postmenopausal osteoporosis treatment is the bisphosphonate zoledronate [172,173]. However, the intake is time limited as the downside of bisphosphonate treatment consists of side effects such as the development of atypical femoral fractures [174], hypocalcemia [175,176], or osteonecrosis of the jawbone [177]. Therefore, drug holidays after at least five years of medication are necessary [178].

#### 3.2.2. Selective Estrogen Receptor Modulators Inhibit the Activation of Osteoclasts

Another well-established group of drugs for osteoporosis treatment are selective estrogen receptor modulators (SERMs). This type of drug is also categorized to the catabolic treatments of osteoporosis, as it inhibits the function of osteoclasts and therefore prevents bone resorption [179]. The effects of SERMs are similar to estrogen. The most prominent and most commonly used SERMs are tamoxifen [180] and raloxifene [181,182], but bazedoxifene [183] and lasofoxifene [184] are also approved. Tamoxifen is used as a drug for the treatment and reoccurrence of breast cancer [185], whereas raloxifene, lasofoxifene, and bazedoxifene are not used for breast cancer treatment but imitate estrogen activity in bone and are therefore used for treating osteoporosis. These drugs have the ability to bind to estrogen receptors and mediate the suppression of RANKL expression and thus the decrease in osteoclast differentiation and activation, as described by Streicher and colleagues [90]. The binding of raloxifene appears to decrease the resorptive activity of osteoclasts by decreasing interleukin-6 expression up to 50% and tumor necrosis factor α expression up to 30% [186]. Both factors are described to play an important role in bone resorption [187]. It was also described that SERMs, in particular raloxifene, increase TGF-β expression and therefore decrease the number of osteoclasts [188]. Nevertheless, SERMs, like other catabolic drugs for osteoporosis treatment, can have considerable side effects. Mainly, SERMs display a low incidence of side effects such as hot flashes and leg cramps, but can also cause blood clots and thus thrombosis [189]. Taken together, the characteristics of SERMs makes them a promising alternative for the treatment of osteoporosis, but there is still space for improvement.

#### 3.2.3. Monoclonal RANKL Antibody Prevents Receptor Activation on Osteoclasts

Monoclonal RANKL antibodies are a third group of catabolic treatments for osteoporosis. The most commonly used RANKL antibody is Denosumab [190]. The antibody binds to RANKL, which is released by osteoblasts. This binding prevents RANKL from binding to its receptor on osteoclast precursor cells and mature osteoclasts. With reduced RANK–RANKL binding, the osteoclastogenesis, activation, and survival of osteoclasts are inhibited, bone resorption is decreased, and thus bone mineral density is increased [191]. When Denosumab was tested in clinical studies, a rapid onset of action, a good tolerability, and a sustained effect for several months was noted [192]. These results were further validated by Cummings and colleagues, who found in a randomized study that Denosumab had a positive effect on postmenopausal osteoporotic women by reducing the fracture risk [193]. However, monoclonal RANKL antibodies also have disadvantages. Since RANKL is abundantly expressed in dendritic cells and activated T lymphocytes [194], Denosumab was shown to affect the immune system and result in adverse side effects for some individuals, such as atypical femur fractures [195] and osteonecrosis of the jawbone [196]. Furthermore, similar to bisphosphonates, monoclonal RANKL antibodies can also cause osteonecrosis of the jawbone [197]. Taken together, Denosumab is one of the most promising treatments for osteoporosis.

#### 3.2.4. Cathepsin K Inhibitors Prevent Type I Collagen Degradation

Cathepsin K is a cysteine protease involved in bone resorption and remodeling [198]. The enzyme is expressed predominantly in osteoclasts and is secreted into resorption lacunae below active osteoclasts [199,200]. After secretion, the cysteine protease actively degrades collagen type I and II fibers of the bone [201,202]. In osteoporosis, bone is excessively resorbed, which is partly based on cathepsin K activity. Cathepsin K inhibitors are therefore a promising new possibility for osteoporosis treatment [203]. Additionally, the inhibition of cathepsin K has one major additional benefit compared to other treatments for osteoporosis. All previously mentioned treatments have adverse side effects and therefore a limited intake duration. These side effects are often caused by the fact that either osteoblast or osteoclast differentiation and activation are affected by the mentioned drugs. This leads to a disturbance of the interplay between osteoblasts and osteoclasts and affects the feedback regulation loop of both cell types. Cathepsin K inhibitors are supposed to inhibit the active form of cathepsin K after it was already secreted by osteoclasts. Therefore, the interplay between osteoblasts and osteoclasts would remain untouched. So far, several cathepsin K inhibitors were tested in clinical trials, but there is still no approved drug. Balicatib terminated in phase II clinical studies [204,205], ONO-5334 passed clinical studies phase I and II and is still under further investigation [206,207], and Odanacatib, the most promising cathepsin K inhibitor which showed a high therapeutic efficacy, was terminated after a clinical trial phase III study showed adverse side effects in non-bone tissue of the osteoporotic patients [208]. As cathepsins are a family of proteases with functions in a variety of tissues, these findings must be expected when the inhibitors are not specific enough for bone tissue. Nevertheless, the approach of inhibiting the protein, which is a key player in bone resorption, is an ideal strategy to treat osteoporosis, but the development of an ideal cathepsin K inhibitor is required to improve the possible outcome.

### 3.3. Combinational Therapies to Improve Osteoporosis Treatment

Nowadays, the most common treatments for osteoporosis are bisphosphonates, monoclonal RANKL antibodies, and sclerostin antibodies, but, as mentioned already, the usage of these drugs is time limited due to adverse side effects [209]. Additionally, although these drugs are approved, there is a general lack of knowledge regarding how to use these drugs effectively. Teriparatide, for example, is a potent anabolic drug for osteoporosis treatment, but medical doctors are unsure if the prescription alone is effective, or if it would be more beneficial to combine the drug with another drug. Therefore, a new approach nowadays is the combination of several drugs or treatment techniques to reduce side effects and to design a permanently effective treatment strategy for osteoporotic patients.

In 2021 in a study from Wei and colleagues, the efficacy of the parathyroid hormone teriparatide alone and in combination with the bisphosphonate zoledronic acid was compared for osteoporosis treatment in postmenopausal women [210]. Ninety-six patients were distributed into two groups. One group was treated with parathyroid hormone 1-34 alone, while the other group was treated with parathyroid hormone 1-34 plus zoledronic acid. After treatment for six months, the group treated with both drugs showed significantly lower levels of bone resorption markers and a higher bone mineral density, indicating that treatment with both drugs was more effective than treatment with parathyroid hormone 1–34 alone [210].

In another study conducted by Shimizu and colleagues, also from 2021, the effects after starting or switching from bisphosphonates to romosozumab, a monoclonal sclerostin antibody, or to Denosumab, a RANKL antibody, in Japanese postmenopausal women was compared [211]. A total of 154 postmenopausal and osteoporotic women were recruited, and their therapy was switched from bisphosphonates or vitamin D to romosozumab or Denosumab. The results of this study showed that the treatment with romosozumab increased BMD for twelve months and therefore performed better than Denosumab. On the other hand, results showed that the switch from bisphosphonates to romosozumab had the same effect on the femoral neck and the total hip if compared to Denosumab. Thus, Shimizu and colleagues concluded that further investigations are necessary, but they also provided useful parameters for predicting the efficacy of romosozumab [211].

Additional studies comparing the treatment of osteoporosis by using several drugs are ongoing. One clinical trial phase II study in the US conducted by Shane and colleagues (NCT02049866) analyzes the effect of Denosumab in preventing post-teriparatide bone loss in postmenopausal women by investigating the effects on bone mass and other osteo-related factors.

Another clinical trial phase II study (NCT03396315) by Shane and colleagues has the goal to assess to what extent bisphosphonate therapy will prevent decreased bone mass that might have occurred after the cessation of Denosumab. Alendronate and zoledronic acid will be used as bisphosphonates after discontinuing Denosumab treatment.

A third example of ongoing studies investigating the combination of several drugs for osteoporosis treatment is being performed by Shoback and Schafer from the San Francisco VA Medical Center (NCT03994172). In this clinical trial phase IV study, novel combination therapies for osteoporosis in men are investigated. One combination will be the treatment with teriparatide, given along with calcimimetic, a drug that activates calcium receptors of osteoblasts and thus activates bone formation. The effect will be compared to the treatment with teriparatide alone. This combination of treatment was already investigated in mice and showed a positive effect by improving the bone mineral density and structure over six weeks of treatment [212,213].

Besides the studies in humans, there are several studies published in animal models where different combinations of drugs are tested for an improved outcome. One combination is zoledronic acid and propranolol, a beta blocker [214]. Another combination of drugs tested is alendronate and alfacalcidol, an active metabolite of vitamin D [215]. Additionally, anabolic agents such as teriparatide and bisphosphonates such as alendronate or zoledronate were combined and have shown to improve BMD [216,217,218]. Therefore, there is already potential for more clinical studies to find an improved drug combination for osteoporosis treatment in the near future. Additionally, treatments of osteoporotic patients already suffered fractures of vertebral and critical size bone defects are targeted with news approaches using biomolecules and biomaterials, which will be introduced and discussed in the next chapter.

### 3.4. Treatment of Vertebral and Critical Size Fractures of Osteoporosis Patients Using Scaffold Material and Controlled Drug Release

#### 3.4.1. Scaffolds Used in Osteoporosis Treatment

Since in vitro cell growth is limited to two dimensions, stem-cell-based tissue engineering requires tailored scaffold materials to guarantee a three-dimensional migration and ingrowth of the stem cells into the scaffold bulk. Today, scaffold-based tissue bone engineering is an essential approach in osteoporosis therapy [219,220,221,222]. Particularly in the case of critical size fractures, scaffolds are designed to provide an environment that mimics the natural origin. Scaffolds used in stem-cell-based therapies can improve the osteogenic differentiation and hard tissue formation. The most important scaffold characteristics include the porosity and pore sizes of the scaffolds to allow cell attachment (adhesion), ingrowth, proliferation, and osteogenic differentiation. Furthermore, the diffusion of nutrients and oxygen as well as metabolite dissipation depend on scaffold porosity.

In general, scaffold characteristics can be divided into two groups (Figure 3) [222]:i.Bulk properties: including porosity/pore sizes, mechanical strength (stiffness and flexibility), biocompatibility, and last but not least, the (bio)degradability of scaffold components and metabolic products. In summary, these parameters are confirmed to guide the 3D cell migration, proliferation as well as final osteogenesis. Moreover, scaffolds’ development also should support angiogenesis, e.g., providing a tailored porosity for the ingrowth of cardiovascular tissue. Finally, the scaffold should be biodegradable in order to allow natural bone growth. Therefore, the products of this degradation are required to be non-toxic and non-mutagenic.ii.Surface properties: including hydrophilicity versus hydrophobicity due to corresponding functional groups (polar or apolar, charged or neutral chemical groups) attached to the surface and resulting in a certain wettability behavior; surface architecture including roughness and topography. For example, scaffold surfaces can be modified to allow a certain cell alignment. Function of the tailored surface include mimicking the surface of the ECM, guiding the cell adhesion, and final cell attachment.

Both bulk and surface of the scaffolds can be tailored using certain fabrication methods. The final scaffold geometry and shape can be tailored by the help of modern additive manufacturing methods. Today, three-dimensional porous scaffolds can be produced that possess shape memory effects for bone tissue engineering. Both in vitro and in vivo studies revealed a significant influence of scaffold order and symmetry onto cell differentiation and proliferation [223,224,225,226,227]. Although, the systemic effect of the immune system can reject the biomaterial and that could lead to failure in vivo, this effect is not further considered and discussed. Here, some of the most important techniques should be briefly discussed. These methods are used in modern tissue engineering approaches for both scaffolds and drug release material manufacturing [228,229,230]. The following techniques are the best known:–Chemical and physical vapor deposition (CVD and PVD): used for surface functionalization and modification; –Self-assembly methods, applied to cover scaffold surfaces with polymer mono- and/or multilayers to modify the surface chemistry and/or topography in a very controlled manner, e.g., layer-by-layer (LbL) method and the Langmuir–Blodgett technique; –Electrospinning of fiber-based scaffolds that mimic the extracellular matrix. Nano-scaled fibrillary structures with diameters of 50–500 nm are accessible—enhancing the cell–matrix interactions and calcification. Increasing the length of electrospun fibers (via polymer concentration) results in enhanced fiber entanglements, thereby improving the osteoconductive activity; –Solid freeform rapid prototyping including selective laser sintering (SLS), selective laser ablation (SLA), and fused deposition methods (FDMs) are used for scaffold surface patterning in different micro- and nano-scales. FDMs allow easy and flexible material implementation for scaffold fabrication reported for bone tissue engineering [228,231]; 3D printing and 3D plotting combines rapid prototyping technologies to produce tailored 3D scaffolds. Thus, scaffold size, shape, and porosity can be adjusted. Interconnected pores are accessible supporting cell ingrowth, metabolic activity, and nutrient exchange [232,233,234,235]; –Lithographic methods can be used, and finally;–Chemical nano-structuring methods such as multiblock copolymers [229,236,237].

The scaffold pore sizes have to be as large as required by the stem cell size to support their migration but also small enough to ensure cell binding to the biomaterial. Further-more, processes such as tailored scaffold degradation and tissue regeneration must be in optimal balance. Beside this, a number of further important criteria have to be considered when designing bone scaffolds: biocompatibility as well as non-toxicity of original scaffold materials and corresponding degradation compounds. In addition to osteogenesis, a high porosity is also required to allow angiogenesis and controlled drug delivery (see also Section 3.4.2).

Scaffolds for bone tissue engineering can be divided into pure inorganic materials (based on hydroxyapatite, HA), natural and synthetic polymers, and composites (hybrid materials) such as advanced ceramics, which have been designed to increase tissue interactions. In Table 2, their advantages as well as disadvantages are summarized (Table 2) [222,238,239].

The first generation of biomaterials showed sufficient biocompatibility but lacked bio-interactivity, so the next-generation materials such as porous composites and functional coatings for metallic implants were developed. The third generation of biomaterials possess a bio-responding feature, which is the ability to activate genes for controlled proliferation and stem cell differentiation. To do so, scaffold materials are combined with options for a controlled drug release [227,228,239,252].

In conclusion, scaffolds for osteoporosis treatment are currently used to ensure a certain mechanical stabilization but also to supply specific drugs. Here, the scaffold acts as a substitute for a separate drug delivery material, as discussed in the next section.

#### 3.4.2. Controlled Drug Release Used in Osteoporosis Treatment

As discussed in Section 3.1, the currently available osteoporosis drug therapy can be classified as osteo-anabolic, catabolic, or dual effective according to their predominant mechanism of action [254,255]. Today, the following recommendations on the duration of osteoporosis medication are reported as guidelines developed by the German Association of Osteology. So, a specific osteoporosis therapy should be re-evaluated after three to five years of therapy with regard to benefits and risks. Here, the personal situation and additional diseases or a changed life situation of those affected must be taken into account [172,173,195,256,257,258]:Teriparatide therapy should generally be limited to 24 months;Raloxifene therapy has been shown to be beneficial for up to eight years;Bisphosphonate therapy has proven benefits for three to five years;Denosumab has a proven benefit for up to three years.

Future developments in osteoporosis treatment should also include the encapsulation and kinetically controlled release of the above-mentioned drugs. So far, a broad variety of materials have been studied to be used for drug encapsulation and controlled release. Detailed knowledge on appropriate release systems is available from stem-cell-based osteogenesis reported by the group of Tobiasch and colleagues [259,260,261].

The most recent study reports the kinetically controlled and sustained release of a purinergic receptor P2X7 antagonist using a lignin-derived layer-by-layer release material [262]. Table 3 gives an overview of the most recently reported materials, including inorganic compounds, polymers, inorganic/organic hybrid systems as well as polymeric hydrogels. A special focus is directed toward nano-scaled materials used for osteoporosis treatment. Today, nanomaterials include different types such as gels and hydrogels, spheres, tubes, and other particles as well as nano-based micelles involved in preclinical and clinical studies. However, further intensive research is required, particularly regarding the reproducibility of the manufacturing techniques and scale-up methods as well as the toxicity of nanomaterials and drug-loading capacity [229,251,252]. In conclusion, previous studies (in vitro and in vivo) could confirm that a controlled drug release is a promising approach for osteoporosis treatment. Both the osteogenesis and even angiogenesis of mesenchymal stem cells can be enhanced, resulting in improved bone healing effects. The most promising examples combine scaffold function with controlled drug release. Future studies have to confirm that this approach might be one of the most promising for bone regeneration in osteogenic therapy.

## 4. Future Perspectives of Anabolic and Catabolic Treatments for Osteoporosis

Treatment options for osteoporosis are progressing promisingly as more and more drugs are approved. Nevertheless, there is still room for more improvements, considering adverse side effects and the lack of knowledge regarding how to use the drugs effectively [209]. Further studies regarding multiple drug therapy must be performed, and detailed research on the mechanisms of action of the drugs is necessary to better understand the molecular basis of the mechanisms and predict and avoid side effects more efficiently.

Beside already established treatments for osteoporosis, novel yet unapproved drugs are promising for future therapy approaches. The mentioned cathepsin K inhibitors are of special interest, as such a potential therapy does not interfere with the feedback regulation between osteoblasts and osteoclasts as most other therapies do, but rather solely targets the secreted protein mainly responsible for bone resorption. However, cathepsins have roles all over the body. Thus, such a receptor must be very specific. Studies with modified cathepsin K inhibitors are already ongoing [272,273,274], and auspicious outcomes are expected.

Another cell type that plays a role in the homeostasis of bone which was not focused on in this review are osteocytes. Osteocytes derive from osteoblasts and are also bone-forming cells, arranged in the bone-remodeling process. A recent review from Pathak and colleagues summarized and discussed the potential of the anabolic treatment of osteocytes to improve bone growth in osteoporosis patients [275]. This is in line with approaches to treat osteoporosis with sclerostin antibodies, as osteocytes produce sclerostin. However, aside from that, no treatments regarding osteocytes or their impact on bone homeostasis were developed so far. As Rochefort and co-worker speculated already in 2014, which was supported by Pathak and colleagues in 2020, future research targeting osteocytes might lead to the development of promising drugs against osteoporosis [275,276].

Other encouraging approaches to treat osteoporosis include targeting specific genes that have an impact on bone homeostasis such as Runx1, a central regulator for osteogenesis [38], the use of epigenetic enzyme inhibitors [277], or the development of RNA-based therapies with mRNAs, siRNAs, and miRNAs [278]. It has been summarized by Ultimo and colleagues that miRNAs, for example, play an important role in the control of skeletal muscle regeneration and function and therefore have an impact on bone tissue [279]. It has been reported that different physical activities modulate different miRNAs which lead to a different expression of genes. Thus, the understanding of miRNAs’ regulation during different types of exercise, regeneration phases, or the aging process in general might help to develop an miRNA-based therapy for osteoporosis [279]. Furthermore, diseases related to osteoporosis, such as sarcopenia, have also been investigated towards the miRNA pattern [280]. Cannataro and colleagues recently summarized the etiology, nutritional approaches, and miRNA background of sarcopenia, a muscle disease based on adverse muscle changes [280]. The disease is leading to a decrease in motoric abilities and therefore to multiple falls which can easily cause fractures in osteoporosis patients. Studies investigating diseases related to osteoporosis are therefore necessary and can lead to approaches for treating osteoporosis.

Another important point to consider in the light of recent years is how osteoporosis interferes with coronavirus disease 2019 (COVID-19). Tsourdi and colleagues addressed this question, and could show that osteoporosis is not associated with a higher risk of COVID-19 infection. Further, it was shown that osteoporosis treatments do not interfere with the efficacy of any COVID-19 vaccination, but possible long-term side effects, although not very likely, such as cross-reactivity of the drugs, have not been evaluated yet [281]. Since the importance of physical activity and exercise to prevent fractures in osteoporotic patients was previously established, it is also important to address the question of how osteoporosis patients might be endangered due to quarantine, staying at home, and movement restrictions [282]. It has been shown that an epidemic group, which was restricted in movement, showed a significantly higher rate of osteoporotic fractures [283]. Additionally, increased telemedicine consultations were observed, DXA scans got delayed, and the supply of medication was interrupted [284]. These findings demonstrate that, although relevant medications and the disease itself do not interfere with one another, osteoporotic patients could be significantly affected by lifestyle changes and restrictions associated with COVID-19 [283]. All in all, we can summarize that the treatment of osteoporosis is complex and has many facets. A drug-based treatment is necessary since lifestyle approaches alone cannot prevent or cure osteoporosis. The commonly used bone-formation-promoting anabolic treatments such as the parathyroid hormones teriparatide and abaloparatide as well as sclerostin antibodies such as romosozumab show an increased bone formation, while on the other hand, catabolic treatments such as bisphosphonates, selective estrogen receptor modulators, and RANKL antibodies show a decreased bone resorption. Both types lead to an increased bone mass and reduce the risk of fractures. Additionally, both types of drugs have similar downsides resulting in adverse side effects and therefore in a limitation on the duration of administration. A promising alternative could be the switch from one drug to another whenever a drug holiday is necessary. Since the two types of medication address different cell types, it would be beneficial to switch from an anabolic to a catabolic drug and vice versa. Shimizu and colleagues tried different approaches by changing from a catabolic treatment with bisphosphonates to an anabolic treatment with sclerostin antibodies, comparing it to changing from a catabolic treatment with bisphosphonates to another catabolic treatment with RANKL antibodies [211]. The strategy of using the same type of drug treatment was not as beneficial for the BMD as if the drug type was changed [211]. A similar approach is currently under investigation by Shane and colleagues (NCT02049866). The switch after treatment with the anabolic drug teriparatide (PTH) to a treatment with a catabolic drug Denosumab (RANKL antibody) is compared to the switch of the catabolic treatment with bisphosphonates to Denosumab. The results will be expected in the near future and are anticipated so that they can be compared with the results of Shimizu and colleagues. Moreover, there are multiple other possible combinations of treatments that need to be investigated. Therefore, it will be interesting and promising to see which combination of anabolic and catabolic drugs could be used for osteoporosis treatment in the future. Given the strengths of the approach of Shimizu and colleagues, it is very likely that a catabolic treatment with bisphosphonates followed by an anabolic treatment could be one of the most promising approaches.

## Figures and Tables

**Figure 1 ijms-23-01393-f001:**
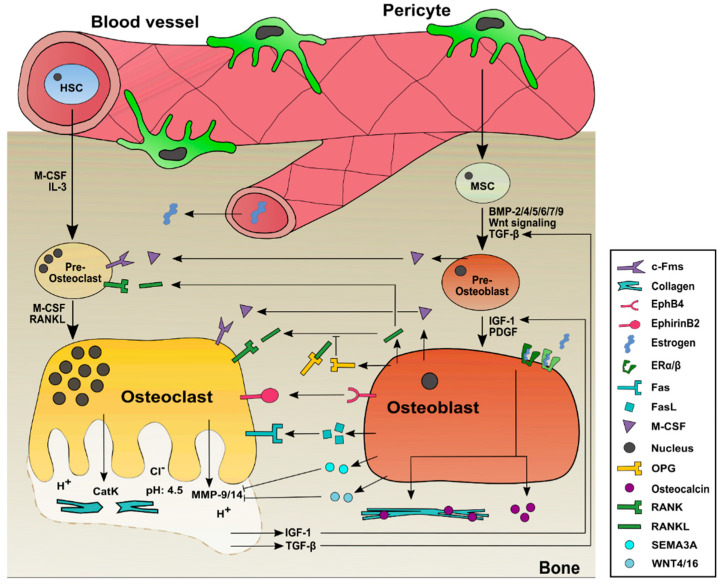
Bone formation, activation, and differentiation of osteoblasts and osteoclasts in healthy individuals. Bone-resorbing osteoclasts derive from hematopoietic stem cells (HSCs) with an intermediate state of pre-osteoclasts. Important factors for the differentiation of HSCs towards osteoclasts are M-CSF, interleukin-3, and RANKL. Mature osteoclasts release cathepsin K and MMPs at the ruffled border into the sealing zone where bone is resorbed and factors such as insulin-like growth factor 1 and TGF-β are released. These factors are needed for osteoblastogenesis. Mesenchymal stem cells, derived from pericytes, need BMP-2/4/5/6/7/9, Wnt signaling, and TGF-β to differentiate towards pre-osteoblasts, followed by insulin-like growth factor 1 and platelet-derived growth factor release, which are necessary to form mature bone-synthesizing osteoblasts. Estrogen is necessary for the differentiation and activation of osteoblasts as well. It binds to its estrogen receptors-α/β (ERα/β) and activates collagen 1 and osteocalcin production in mature osteoblasts. Both cell types play a major role in bone homeostasis and exchange factors to activate each other. RANKL and M-CSF are produced by osteoblasts and are needed for the differentiation and activation of osteoclasts. OPG is also produced by osteoblasts but is a decoy for RANKL, therefore inhibiting osteoclast activation and differentiation. FasL, ephirins, semaphorins, and WNTs also play a role in cell communication between osteoblasts and osteoclasts, as indicated.

**Figure 2 ijms-23-01393-f002:**
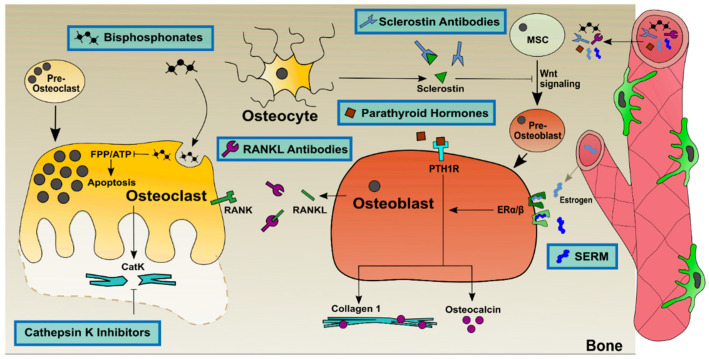
Overview of the different anabolic and catabolic treatments on osteoblasts and osteoclasts for osteoporosis. Bisphosphonates act on adenosine triphosphate and on farnesyl pyrophosphatase and lead osteoclasts into apoptosis. RANKL antibodies bind to osteoblast-produced RANKL and prevent RANKL from binding to RANK on osteoclasts and therefore inhibit activation and differentiation of osteoclasts. Cathepsin K inhibitors inactivate the mature cysteine protease cathepsin K and prevent bone resorption. Sclerostin antibodies bind to sclerostin that is produced by osteocytes, inhibit Wnt signaling and therefore osteoblast differentiation. When sclerostin antibodies are present, osteoblastogenesis is active. SERM are similar to estrogen and can bind to the estrogen receptors α/β, which leads to an activation of osteoblasts.

**Figure 3 ijms-23-01393-f003:**
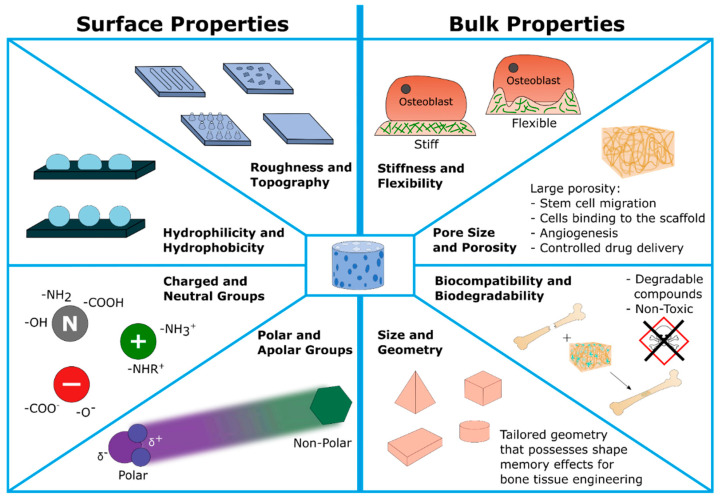
Overview of bone scaffold characteristics. Including: scaffold size and geometry, mechanical strength (stiffness and elasticity), geometry, surface hydrophilicity/hydrophobicity, surface charge, pore size, porosity, and others, modulating the host responses and bone regeneration.

**Table 1 ijms-23-01393-t001:** Risk factors for osteoporosis. Risk factors can be separated into multiple groups. The first risk factor group is the age-related loss of sex steroids and hormonal changes, while another group of risk factors include environmental and other external factors. All groups were separated into subgroups to depict which factors increase the risk of osteoporosis and how.

Risk Factor Group	Risk Factor Subgroup	Effect/Influence	References
Hormone reduction	Estrogen/Estrone reduction	Reduction in OPG expression	[89]
		Lack of ER-α-mediated suppression of RANKL expression	[90]
		Lack of growth factor production such as IGFs or TGF-β	[91,92]
		Lack of suppressive effect on Wnt-signaling antagonist sclerostin	[93]
	IGF-1/2 reduction	Reduction in osteoblasts’ activation and differentiation	[117]
Medical disorders and medication	Cancer/Breast cancer	Estrogen can influence breast cancer and treatment with anti-estrogen drugs can cause osteoporosis	[98,99]
	Rheumatoid disorders	Glucocorticoid treatment in rheumatoid disorders increases the risk of osteoporosis	[100,101]
		Systemic inflammations leading to bone erosion due to a local effect of immune cells	[100]
	Chronic kidney disease	Therapeutic drugs against osteoporosis can affect the renal function	[103]
	Chronic liver disease	Bilirubin and bile acids are retained factors of cholestasis and can decrease bone formation	[104]
	Diabetes mellitus	Hyperglycemia can be damaging to bone, since glucose can be toxic to osteoblasts	[105,106]
	Parkinson´s disease	Reduced mobility can cause reduced bone mass	[107]
	Multiple myeloma	Changes in the bone marrow microenvironment can lead to a dysregulation of bone turnover	[108]
Poor nutrition/dietary factors	Low calcium intake	Calcium is an essential nutrient for bone growth, and a low intake reduces the bone density	[111]
	Eating disorders/Anorexia nervosa	Low body weight can induce bone loss	[109]
		Gonadal function is decreased and can cause reduced bone mass	
		Metabolic disorders such as growth hormone resistance, low leptin concentrations and hypercortisolemia can induce bone loss	
Lifestyle choices	Sedentary lifestyle	Reduced mobility can cause reduced bone mass	[112]
	Excessive use of alcohol	Alcohol decreases the absorption of calcium and vitamin D	[113]
		Alcohol slows the bone turnover down	[114]
	Use of tobacco	Indirect effect: the alteration of parathyroid hormone, adrenal hormones (leading to hypercortisolism), gonadal hormones, and increased oxidative stress	[113,115]
		Direct effect: binding of nicotine to its receptor on osteoblasts and inhibiting proliferation	[116]

**Table 2 ijms-23-01393-t002:** Current scaffold materials used in osteoporosis treatment with their advantages and disadvantages.

Scaffold Type	Chemical Composition	Advantages	Disadvantages	References
Noble Metals	Titanium (Ti) and corresponding Ti alloys	Inertness, good biocompatibility, high mechanical strength; hydrophilic surface with reduced macrophage activation providing anti-inflammatory microenvironment improving osteogenesis	Strictly limited flexibility, no inherent bioactivity	[223,240]
	Gold (Au) prepared as nanoparticles	Gold nanoparticles available in varying sizes (about 10–70 nm) with specific nano-topography that guides the cell attachment; gold with influence on expression of cytokines as well as different factors (e.g., osteogenic, fibrogenic, and angiogenic factors)		[238,241]
Minerals and Ceramics	Tricalcium phosphate (TCP); hydroxyapatite (HA)	Chemical composition with similarity to native bone tissue, high tensile strength	Low compressive strength and comprehensive modulus; limited option to change surface chemistry (number/nature of functional groups)	[235,242,243]
	Sr–Ca–Si-doped, HA-based scaffolds	Doping enhances biomineralization capacity (compared to non-doped scaffolds) resulting in increased osteogenic activity		[219,239,244]
Hydrogels based on inorganic minerals and/or organic/polymeric materials	Poly(anhydride)s layers loaded with drugs (e.g., teriparatide)	Hydrogels possess a porous structure in micro- and nano-scale for tailored cell adhesion; option for surface modification/functionalization; high tensile strength, in situ formability and in situ drug delivery, injectability; high targeting and ability to allow uniform incorporation of therapeutic molecules and cells, resp., without need for further surgery; good biocompatibility and biodegradability resulting in fewer side effects, good cell colonization and proliferation	Low mechanical/compressive strength; need to combine hydrogels with other components in order to meet mechanical requirements; fast degradation (in case of poly(anhydrides)	[228]
	Bioinspired mineral HA-based hydrogels as nanocomposite scaffolds for the promotion of osteogenic marker expression and the induction of bone regeneration in osteoporosis			[219,245]
Scaffolds based on synthetic polymers	Poly(caprolactone) (PCL)	Proven biocompatibility, option to tailor chemical composition and 3D structure of bilk polymers and surfaces to reach high porosity and tunable pore sizes	Low mechanical strength, premature degradation	[238,246]
	Poly(vinyl alcohol) (PVA) modified with gelatin			[234]
	Polyesters such as poly(lactic acid) (PLA), poly(glycolic acid) (PGA), and copolymers (PLA/PGA)			[231,247,248]
	Poly(ethylene glycol diacrylate) (PEGDA) combined with laponite nanoclay (a mineral consisting of magnesium (Mg), lithium (Li), and silicon(Si))			[249,250]
	Polyetherketoneketone scaffold with a functionalized Sr-doped nano-HA coating	Biomimetically hierarchical structures; due to local release of Sr, enhancement of the intrinsic mechanical strength at microlevel resulting in improved bonding strength (of scaffold and host bone)	Advanced synthetic approach, hydrophobicity of the majority of synthetic polymers	[227]
Scaffolds based on natural polymers	Polypeptides, such as collagen-based scaffolds (including different types I-V, focus on collagen type I); micro/nanoporous collagen modified with silk-fibroin	Collagens (combined with HA) enable the mimicking of chemical composition of natural bone	Weak mechanical strength requires a combination of collagen with minerals such as HA to mimic the chemical composition of bone	[251,252]
	Polyesters such as poly(hydroxy alkanoates) (PHA), poly(hydroxy butyrates) (PHB), and poly(alginates)	Polyesters enable the mimicking of the natural tissue such as ECM; PHA show good biocompatibility and biodegradability, exhibit good tensile strength, thermoplasticity and elastomeric nature	Advanced preparation methods of PHA, PHB using enzymes/bacteria	[253]

**Table 3 ijms-23-01393-t003:** Most recently published examples for drug encapsulation and controlled release used in osteoporosis treatment.

Release Material	Chemical Composition	Encapsulated/Released Drug	Release Results	References
Inorganic/organic hybrid materials	Drug-functionalized HA combined with biodegradable collagen microspheres	Dual release of bisphosphonates (Alendronate, ALN) combined with BMP-2	Initial release of BMP-2 for a few days, followed by sequential ALN release after two weeks; finally, increased osteogenic activity was observed due to synergistic effect of BMP-2/ALN and enhanced bone regeneration at eight weeks post-implantation (rat 8 mm critical-sized defect).	[252]
	Thermo-sensitive triblock copolymer depot for treatment of osteoporosis	Salmon Calcitonin (sCT)	In vitro and in vivo studies using injectable depot materials doped with sCT to prevent osteoporosis side effects; moreover, the copolymeric release system maintained sCT in a conformationally stable form for the entire release process.	[236]
Hydrogels based on pure polymers and/or hybrid materials	Methylacrylated gelatin	Abalo-paratide (analog of parathyroid hormone-related protein PTHrP 1-36)	Controlled release of Abaloparatide via injectable hydrogel—resulting in promoted pre-osteoblast differentiation and final bone regeneration.	[143,218,263]
	PTH—hylaluronic acid hydrogel	Teriparatide (recombinant N-terminal fragment (rhPTH1-34) of the human parathyroid hormone)	Pulsatile drug delivery system, promoting angiogenesis.	[228]
	Collagen-based hydrogels	Sustained delivery of Alendronate	Improved repair of osteoporotic bone defects and resistance to bone loss. Kinetic studies confirmed a sustained ALN release, resulting in repair effects of collagen–ALN scaffolds for osteoporotic defects (5 mm cranial defects in ovary ectomized rats).	[251]
Nanomaterials, e.g., nanoparticles, nanotubes, nanogels	HA-based nanoparticles	Bisphosphonates (ALN, ZOL)	Zn, Sr, and Ag incorporation for invigorating bone growth; partial Ca-substitution with cobalt favor osteogenesis process.	[229]
	Titanium Sr/Ag loaded into Ti nanotubes	Strontium (Sr) and gold (Au)	Controlled drug release by varying nanotube diameter.	[229,264]
	Nanogel scaffolds consisting of bioactive glasses	Strontium (Sr)	Enhancement of osteoblast differentiation; enhanced bone regeneration in osteoporosis rats; targeted therapy of osteoporosis.	[229,265,266]
	Amino modified mesoporous bioactive glass (MBG) scaffolds	Alendronate (ALN)		
	Injectable nanogels consisting of triblock-copolymers	bone-seeking hexapeptide (Asp6)-conjugated sCT (sCT-Mal-Asp6) for the targeted treatment of osteoporosis		[248]
	ZOL-loaded gelatin nanoparticles, integrated porous Ti scaffold implanted in a femoral defect	Zoledronate (ZOL)		[267]
	Mesoporous hydroxyapatite (MHA) modified with poly (N-isopropylacrylamide) (PAA) brushes	Simvastatin (SIM)	Anti-osteoporotic effect of the SIM-loaded PAA/HA system studied in vivo on femur defect.	[239]
	Calcium sulfate/nano-HA-based nanocomposite carrier of BMP-2 and ZOL	BMP-2 and Zoledronate (ZOL)	BMSC-derived EXO, implanted in a femur defect (Sprague Dawley OVX rats) resulting in improved osteogenesis of the BMSCs.	[268]
	Nano-HA, nCh/HA, and nAg/HA delivered intravenously to female albino Wistar OVX rats	Alendronate (ALN)		[269]
	Calcium citrate homogenized to nanoparticles (NPs), combined with PLA- and PLGA-based NPs	17-beta-estradiol	Hormone replacement therapy; polyurethane nano-micelles deliver mRNA.	[229,270,271]

## Data Availability

Not applicable.

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
