# Peer review of "Therapeutic Treatments for Osteoporosis—Which Combination of Pills Is the Best among the Bad?"

_ijms, 2022, doi:10.3390/ijms23031393_

Round 1
Reviewer 1 Report
The article from Edda Tobiasch´s group summarizes the current state of knowledge on therapeutic treatments for osteoporosis – very important topic both for research and the clinics of the musculoskeletal system. The article is very comprehensive and on one side includes findings that have previously been covered but on the other side adds novel ideas and insights as well. The article is well and carefully written and includes nice tables which provide a good overview on those aspects that in parallel are covered in more detail in the accompanying text. In addition, the graphics are well conceptualized. In summary, I really enjoyed reading their careful presentation of state-of-the-art and only have some minor comments.
- Line 43: What is “a special type of mesenchymal stem cells”? The authors can just omit “a special type of” to avoid any discussions about the nature of MSCs.
- Line 932: The sentence “Calcium and vitamin D are not considered part of a combination therapy” is somewhat unmotivated. Either removed it or give a short explanation.
- Figure 1: Please add in the legend what the blue, orange, dark red, and green circles within the major cell types signify. Also, the authors might consider to omit the captions within the figure where symbols are explained in the legend, to make the figure even clearer.
- The authors might consider to add “immune cells” in figure 3. Surprisingly, many studies and consequently many reviews do not consider at all systemic effects of biomaterials including their interaction with the immune system – a very important aspect that leads to failure of many biomaterials in vivo, even when locally implanted.
- There are some minor spelling or English grammar mistakes in the text which might be corrected by MDPI staff during the proofreading stage. In particular, this applies to the title for which I suggest to replace “Worst” by “Worse” or “Bad”.
Reviewer 2 Report
In this review, Tonk and colleagues attempt to summarize ongoing clinical treatments for osteoporosis (OP). Given the high incidence and prevalence of OP, the topic of the manuscript is of high medical interest. I would like to congratulate the authors for explaining well the molecular mechanisms underlying OP and for compiling all the therapeutic approaches currently used to treat OP. The figures and tables are very well structured.
However, the manuscript should be thoroughly revised and restructured to meet publication standards.
Major changes:
The paper needs an introduction that briefly describes:
- The pathogenesis and risk factors associated with OP;
- The prevalence of OP and the different types of OP (as in section 2.2):
- The different classes of drugs currently in use.
Paragraphs 2.1 and 2.2 should be deleted because they do not contribute substantial information to the objective of the article. The information about the different types of OP should be moved to the introduction.
Paragraph 2.3 should be summarized in one figure or shortened and moved to the "State of the Art Treatments Against Osteoporosis" chapter.
Lines 790-823: Techniques for making scaffolds are not useful for the purposes of this review and should be removed or shortened.
Lines 845-884: The authors should remove the information already included in Table 2.
Lines 886-909 are not clearly structured and should be reworded and moved to the discussion section.
Lines 910-960 should be moved to Table 3. Also, the information on side effects in the previous paragraphs should be deleted.
In the last paragraph, the information presented by the authors throughout the manuscript needs to be better discussed. Also, the take-home message and the authors' point of view are missing throughout the paper. What is the answer to the question, "Which combination of pills is the best among the worst?".
Lines 1016-1020 should be deleted. I cannot understand why this information is reported.
Minor changes:
Line 44: BM should be stated.
Lines 71-72: "Differentiation of MSCs begins with the commitment of MSCs to a particular cell line, followed by differentiation"; "followed by differentiation" should be deleted.
Line 85: MSCs from bone marrow can be eliminated. The acronym BM -MCSs has already been shown in line 44.
Line 96: "bone morphogenetic protein 7 (BMP-7)" should be bone morphogenetic protein(BMP)-7.
Line 463: Perhaps "compared to RANKL" should be "compared to RANKL antibody".
Reviewer 3 Report
This paper presents an interesting update on the treatments for osteoporosis.
In my opinion, it should be reduced both in some paragraphs, especially the initial ones, which at times are heavy to read.
Although interesting paragraph 4 (indeed perhaps the most innovative part of the whole work) is detached from the rest, so I would suggest eliminating it, thinking, perhaps, of considering it for a separate review.
Being a narrative review (not systematic) it would be necessary to give more emphasis to the authors' ideas, proposing possible strategies on the basis of the bibliography (perhaps a little abundant) proposed.
Physical activity is simply mentioned, but also in support of therapies (for example 10.1016 / j.smhs.2021.02.005) one should at least consider, in particular, that with resistance training, also because the bone also behaves as organ secretory in response to the exercise.
Finally, the authors cite the interesting field of miRNAs, which should be explored, in relation to physical exercise
(10.18632 / oncotarget.24991) and in the absence of this also to sarcopenia (10.3390 / ijms22189724)
Round 2
Reviewer 2 Report
I must congratulate the authors for their efforts to improve the manuscript. The manuscript is now acceptable and I am sure it can be of help to many readers in the field. It can be accepted without further changes.